# Employing Manganese Dioxide and Bamboo Carbon for Capacitive Water Desalination and Disinfection

**DOI:** 10.3390/nano14191565

**Published:** 2024-09-27

**Authors:** Cuihui Cao, Xiaofeng Wu, Yuming Zheng, Lizhen Zhang, Yunfa Chen

**Affiliations:** 1State Key Laboratory of Multiphase Complex Systems, Institute of Process Engineering, Chinese Academy of Sciences, Beijing 100190, China; caocuihui122@163.com (C.C.);; 2Department of Chemistry and Pharmacy, Guilin Normal College, Guilin 541119, China; 3Center for Excellence in Urban Atmospheric Environment, Institute of Urban Environment, Chinese Academy of Sciences, Xiamen 361021, China; 4University of Chinese Academy of Sciences, Beijing 100049, China

**Keywords:** manganese dioxide, bamboo carbon, CDI

## Abstract

A manganese dioxide (MnO_2_)/bamboo carbon (BC) composite was prepared using hydrothermal and impregnation methods and used for the capacitive desalination (CDI) and disinfection of water. The results showed that these composites had fast Na^+^ ion exchange and charge transfer properties. During the CDI process, these composites’ electrodes exhibited good cycle stability and electrosorption capacity (4.09 mg/g) and an excellent bactericidal effect. These carbon-based composite electrodes’ bactericidal rate for *Escherichia coli* could reach 99.99% within 180 min; therefore, they had good performance and are a good choice for high-performance deionization applications.

## 1. Introduction

With the advancement in society, the visibility of environmental issues has grown, and the scarcity of fresh water has become more pronounced [1,2,3,4]. To tackle this, recycling water resources is essential, a process that necessitates the utilization of water treatment technologies. Presently, the standard methods for water treatment encompass precipitation, filtration, chemical treatment, and advanced oxidation processes. Yet, these conventional methods are not without their drawbacks, as they can result in the creation of toxic byproducts and carcinogens that pose a threat to human health [5,6,7,8]. For instance, by-products of chlorination and disinfection are known to be carcinogenic [9,10,11]. Consequently, there is a pressing need for the development of innovative water treatment technologies.

Capacitive deionization (CDI) represents an emerging water treatment approach, predominantly employed in the initial phase of desalination processes. As the technology has evolved, it has been progressively implemented in the treatment of wastewater, removal of heavy metals, and eradication of bacteria. Characterized by its non-toxicity and absence of pollution, CDI technology offers a simple and cost-effective operation [12,13,14,15]. As compared to surface-based physicochemical methods, CDI providing faster kinetics is one key merit. Secondly, during the CDI process, the presence of an electric field could accelerate the diffusion of ions, making it possible to overcome the adsorption–desorption equilibrium encountered in physicochemical adsorption that is controlled by thermodynamics [16]. However, existing CDI technologies for water treatment still face challenges when dealing with large volumes of water, especially in enhancing the adsorption capacity and rate of electrode materials [17,18].

Mesoporous nanocarbon materials, a type of pyrolyzed carbon, have a low graphitization degree, poor conductivity, and complex preparation processes, which are costly. These shortcomings limit their application in large-scale electrode preparation. Therefore, high-temperature pyrolyzed bamboo carbon, which is widely available and easily sourced, is used as the basic carbon material. By loading metal oxides (RuO_2_, MnO_2_, TiO_2_), the porosity and specific surface area are improved, enhancing the capacitive characteristics of the electrode and the electrostatic adsorption capacity for salt ions [19,20,21,22]. Although existing studies have enhanced CDI performance by using various carbon-based materials and metal oxides, these materials often have issues such as complex preparation, high cost, or poor stability. The synergistic effect in both capacitive deionization and disinfection has not been fully investigated. Currently, how to use sustainable, low-cost materials to improve the efficiency and stability of CDI systems, while achieving the desalination and disinfection of water, remains a significant research gap in this field.

This study innovatively combined manganese dioxide (MnO_2_) with bamboo-based activated carbon to prepare a new type of CDI electrode material. Bamboo-based activated carbon, due to its unique pore structure and high specific surface area, is considered a promising candidate for capacitive water desalination and disinfection [23,24]. Through acidification treatment, we had introduced a large number of oxygen-containing functional groups onto the surface of the carbon, enhancing the hydrophilicity of the entire composite, while MnO_2_ nanoparticles provided sufficient ion migration channels, thus achieving high adsorption capacity and rapid reaction kinetics. The purpose of this study was to synthesize an acidified bamboo-based activated carbon/manganese dioxide (BC@MnO_2_) composite through a simple co-precipitation technique and evaluate its bactericidal performance in CDI. We expected this composite material to demonstrate an excellent specific capacitance, electrosorption capacity, bactericidal effect, and recyclability.

## 2. Materials and Methods

### 2.1. Materials

All chemicals, including potassium permanganate (KMnO_4_, 98%), hydrochloric acid (HCl, 99%), bamboo carbon (BC, 20 nm), ethanol (C_2_H_5_OH, 95%), N-Methyl pyrrolidone (NMP, 99.5%), N,N-Dimethylformamide (DMF, 99%), acetone (CH_3_COCH_3_, 99.5%), acetic acid (CH_3_COOH, 99.5%), potassium hydroxide (KOH, 90%), polyvinylidene fluoride (PVDF, 99.5%), and sodium chloride (NaCl, 99.5%) were purchased from Sigma-Aldrich, St. Louis, MO, USA, and were of an analytical grade.

### 2.2. Preparation of BC@MnO_2_-X

Firstly, the purchased 3000-mesh bamboo carbon (BC) was cleaned with deionized water to remove dust and impurities, then added to 3 M HCl and stirred for 8 h to remove residues, and then cleaned with anhydrous ethanol and deionized water and filtered until neutral, and placed in a vacuum drying box at 80 °C for 4 h. Secondly, the preparation of BC@MnO_2_ is based on the following reaction:4KMnO_4_ + 3C + H_2_O = 4MnO_2_ + 2KHCO_3_ + K_2_CO_3_

In total, 0.36 g of bamboo carbon powder and 6.32 g of KMnO_4_ are uniformly dispersed in 100.00 mL of deionized water under ultrasonication for 30 min, then transferred to a reaction kettle and heated at 70 °C, 90 °C, and 110 °C in an oil bath for 90 min, respectively. The reaction process needs to be carried out under magnetic stirring to prevent aggregation and precipitation. After the reaction is completed, the reaction kettle is cooled to room temperature, the product is centrifuged at 12,000 r/min at 20 °C, the supernatant is decanted, and the product is rinsed with deionized water and transferred to a 100 °C constant-temperature drying box for 6 h. The dried product is placed in a tubular furnace and calcined in an argon atmosphere at 300 °C for 6 h, and the calcined product is the bamboo carbon loaded with MnO_2_, named BC@MnO_2_-X, where X is the reaction temperature.

### 2.3. Characterization

SEM images and nitrogen adsorption isotherms of the samples were measured using field emission scanning electron microscopy (FESEM, JSM-7800) (Electronics Japan-Oxford, TKY, Tokyo, Japan) and an ASAP 2020 (Micromeritics) (NSK Ltd., TKY, Tokyo, Japan), respectively. The Brunauer–Emmett–Teller (BET) method was utilized to calculate specific surface areas, pore volumes, and pore size. X-ray powder diffraction (XRD, X’ Pert Pro MPD, Philips, Almelo, The Netherlands) was employed to detect the crystal structure.

### 2.4. CDI Electrode Fabrication

BC@MnO_2_-X, acetylene black, and PVDF (mass ratio of 8:1:1) were added to the mortar. An appropriate amount of NMP was added to fully grind the mortar to obtain a slurry. Graphite paper (5 × 5 cm^2^) was coated with the slurry; samples were then transferred to dry in a vacuum drying oven at 80 °C.

### 2.5. CDI Experiments

The CDI device consisted of four main firmware: a DC power supply, a conductivity monitor, a peristaltic pump, and a CDI cell (Figure 1). The CDI cell included four main parts: end plates, soft silica gel plates, electrodes, and a spacer. The bio-contaminated water (*E. coli* suspension with 10^6^ CFUmL^−1^)/NaCl (200 mg·L^−1^)) was pumped into the CDI device at a flow rate of 12 mL/min under 1.2 V.

### 2.6. Electrochemical Measurements

The electrochemical performance of the samples was evaluated using cyclic voltam-metry (CV), which was performed in a CHI 660D electrochemical workstation using a three-electrode system. The system included a saturated calomel electrode (the reference electrode), a platinum gauze electrode (the counter electrode), and a BC@MnO_2_-X electrode (the working electrode). This experiment was performed in triplicate. The specific capacitances were calculated using the following formula:$$C=\int \frac{IdV}{2v∆Vm}$$
where was the specific capacitance (F/g), *I* was the response current density (A), *V* was the voltage (V), *v* was the potential scan rate (V/s), and *m* was the mass of the electrode material (g).

### 2.7. Preparation of Microbial Cells

*Escherichia coli* (ATCC8739), broths, and agar media were obtained from American Type Culture Collection and Becton Dickinson Company (Franklin Lakes, NJ, USA). Freeze-dried bacteria were inoculated in Mueller Hinton (MH) broth and cultured at 37 °C overnight to recovery. Bacteria cells were inoculated in LB agar, cultured at 37 °C overnight, and then harvested, centrifuged, and washed with a phosphate-buffered saline (PBS) solution to remove residual nutrition. Cell numbers were determined using the plate colony counting method. Next, 100 µL of 10-fold dilutions was pipetted into the LB agar of a disposable sterile culture plate. Plates were cultured in a humidity incubator at a constant 37 °C overnight for colony formation.

### 2.8. In Vitro Culture

First, the BC@MnO_2_-X samples, which were dispersed in sterile water under ultrasonication, were sterilized under a UV lamp for 30 min. Next, 1 mL of 10^6^ CFU cells was pipetted into the dispersion of the sample and cultured in vitro. The in vitro culture condition was 200 rpm shaking at a constant 37 °C. After in vitro culturing for 15, 30, 60, 120, and 180 min, 0.1 mL of each suspension was pipetted into LB agar and cultured overnight at 37 °C.

### 2.9. CDI Percent Killing Calculation

The *E. coli* suspension (10^6^ CFUmL^−1^) was prepared as the starting bio-contaminated water. During the CDI process, 0.1 mL of CDI outflow was pipetted into the LB agar of a disposable sterile culture plate and incubated overnight at 37 °C for colony formation. This experiment was performed in triplicate. The percentage kills were calculated using the following formula:$$\%kill=\frac{cellcountofcontrol-survivorcountonsample}{cellcountofcontrol}\times 100\%$$

## 3. Results and Discussion

### 3.1. Characterization of BC and BC@MnO_2_-X

Figure 2 presents the FESEM images of samples BC and BC@MnO_2_-X. As can be seen from Figure 2a–d, the bamboo carbon material was a solid block of 30–50 μm, with a body that retains the pores of the wood fiber, with a pore diameter of about 3 μm. This through-pore structure was beneficial for the infiltration of the electrolyte and the electrode, which was advantageous for improving the capacitance of the electrode and the adsorption capacity of ions. Figure 2b–d revealed that the inner wall surface of the bamboo carbon’s pores had become rough, indicating that the bamboo carbon BC sample, after being loaded with MnO_2_, affected its surface and pore smoothness. Moreover, as the reaction time increases, the roughness of the surface also increases. However, it was evident that MnO_2_ loaded at 90 °C on the bamboo carbon surface forms a more uniform MnO_2_ nanosheet structure. This stacked nanosheet structure on the surface, upon physical contact with bacteria, could damage the bacterial cell wall or membrane with its sharp edges, whereas the MnO_2_ loaded at 70 °C on the bamboo carbon surface forms a nano-protruding particle structure: the BC@MnO_2_ at 110 °C contains a large number of detached aggregated MnO_2_ particles. Although the particulate MnO_2_ had some bactericidal activity, its effect on contacting microbial cells and inactivating them are limited.

The N_2_ adsorption–desorption isotherms and pore size distribution curves of sample BC@MnO_2_ are shown in Figure 3a,b. According to the physical adsorption isotherm types published by the International Union of Pure and Applied Chemistry (IUPAC), it was found that the N_2_ adsorption–desorption isotherms of samples BC and BC@MnO_2_-X conform to Type I. Table 1 indicates that the specific surface area and pore diameter of BC are 122.50 m^2^/g and 56.70 Å, respectively. After loading MnO_2_ on the sample, significant changes in specific surface area and pore size occurred. Among them, the specific surface area of sample BC@MnO_2_-90 was 192.83 m^2^/g, and the pore size was 51.80 Å. This indicated that loading MnO_2_ could significantly increase the specific surface area and pore volume, significantly improving the interfacial properties of the material. In contrast, the BC@MnO_2_-70 and BC@MnO_2_-110 had a lower specific surface area and pore size, indicating that the granular MnO_2_ formed has a higher density and destroys the micropores of BC. However, the BC@MnO_2_-90, due to the formation of a uniformly stacked MnO_2_ nanosheet structure on the bamboo carbon surface, had produced the porosity of MnO_2_ and new interfaces, which were beneficial factors for the infiltration of the electrolyte on the electrode and the contact adsorption of cells.

The XRD results of samples BC and BC@MnO_2_-70, -90, and -110 are shown in Figure 4. It can be seen from Figure 4 that sample BC had diffraction peaks at 23.13° and 44.28°, corresponding to the (004) and (102) crystal planes of BC, respectively. The diffraction peaks of BC@MnO_2_-70, -90, and -110 were similar, all appearing at 28°, 37°, 41°, 42°, 46°, 56°, 59°, 64°, 67°, and 72°, corresponding to the (110), (101), (200), (111), (210), (211), (220), (002), (310), and (301) crystal planes of β-MnO_2_ (JCPDS No. 24–0735) [25]. This also indicated that BC had been successfully loaded with MnO_2_, and a nanostructure of manganese oxide had been formed, which is consistent with the observation from the FESEM image.

The electrochemical performance tests of the samples are shown in Figure 5, which are the cyclic voltammetry (CV) curves of the samples at different scan rates. It can be seen from Figure 5a–d that the CV curves of bamboo carbon BC and BC@MnO_2_-70, -90, and -110 all had a typical spindle-shaped hysteresis loop characteristic, indicating that as the electrode polarization increased, the charge storage process of the electrode changes from a typical “pseudocapacitance” characteristic to being dominated by the dynamics of the electrolyte diffusion [26]. The current of the electrode changes more obviously with the change of potential, and as the loading temperature increases, the sample loaded with MnO_2_ had a higher specific surface area, and this effect was more obvious. In addition, as the potential scanning speed increased, the diffusion effect of the sample loaded with MnO_2_ after loading was also more obvious, but as the loading temperature increased, the diffusion effect weakens. This was mainly because the MnO_2_ loaded at high temperatures formed a large number of isolated MnO_2_ aggregate particle structures, which were not conducive to ion diffusion. According to the formula, at a scan rate of 100 mV/s, the specific capacitance of BC was 23.52 F/g, and the specific capacitances of BC@MnO_2_-70, -90, and -110 were 26.41 F/g, 33.57 F/g, and 30.49 F/g, respectively. These data indicated that the BC electrode material with a stacked MnO_2_ nanosheet structure on the surface, due to the formation of new interfaces of porous MnO_2_, improved the adsorption of Na^+^ on the electrode; but as the loading of MnO_2_ increased, MnO_2_ tended to aggregate into irregular large particles, which was not conducive to ion adsorption.

Figure 6 shows the conductivity/time variation curves and electrosorptive capacity/time variation curves of BC and BC@MnO_2_-70, -90, and -110 materials during CDI desalination. It can be seen from the figure that in the initial stage, the rate of change of conductivity was relatively fast, indicating that after the voltage of 1.2 V is applied, ions are quickly adsorbed by the electrode material. After the experiment started, the conductivity of the BC electrode material decreased from 541.10 µS/cm to 535.20 µS/cm, and the conductivity of the BC@MnO_2_-90 electrode material decreased from 541.10 µS/cm to 530.80 µS/cm. Loading MnO_2_ could increase the electrosorption effect of the BC electrode material, and the electrosorption effect of the BC@MnO_2_-90 material was better than that of BC@MnO_2_-70 and -110 materials, indicating that the nanosheet structure formed by loading MnO_2_ could enhance the electrosorption effect of the bamboo carbon material, and was better than the electrosorption effect of the nano-aggregated particle structure formed by loading MnO_2_. After turning off the power for reverse electrification and desorption, it was found that the change in conductivity was very obvious in the initial stage of reverse electrification. Due to the different electrosorption amounts of different materials, the material with a better adsorption effect had a longer desorption time, but the overall desorption time was not high, and both BC and BC@MnO_2_ electrode materials showed good stability after completing four cycles, indicating that the electrode material could quickly complete the adsorption and desorption of ions in the CDI process and had good cycle stability.

Figure 6b shows the electrosorptive capacity/time variation curves of the material electrode during CDI desalination. It can be seen from the figure that the electrosorption capacity of the BC@MnO_2_-90 electrode material was higher than that of BC@MnO_2_-70 and -110, reaching 4.09 mg/g, indicating that the nanosheet structure formed by loading MnO_2_ at the appropriate temperature could better enhance the capacitance performance of the material and increase the electrosorption amount of the material.

### 3.2. Antimicrobial Activity

Figure 7 shows the comparison of the number of colony-forming units after 15 min and 180 min of the in vitro culture with BC and BC@MnO_2_-70, -90, and -110 materials. It was observed that there was a significant change in the number of colonies after both 15 min and 180 min of the in vitro culture. Specifically, the BC@MnO_2_-90 material showed a more pronounced reduction in the number of colonies after 180 min of the culture, decreasing from 22 to 4. This indicated that the BC@MnO_2_-90 material had the best bacterial removal effect in the in vitro culture. It was suggested that during the in vitro bacterial culture, the uniform nanosheet structure of MnO_2_ loaded on the BC@MnO_2_-90 material not only enhanced the material’s adsorption properties but also increased its physical cutting effect, thereby improving the material’s bactericidal performance.

Figure 8 illustrates the bacterial killing rates after the in vitro culture with BC and BC@MnO_2_-70, -90, and -110 materials for 15, 30, 60, 120, and 180 min. It can be seen that the BC material had a bacterial removal rate lower than 80.39% after 15 min of the in vitro culture, while the BC@MnO_2_ materials had a killing rate lower than 78.43% after the same period. After 180 min of the in vitro culture, the BC@MnO_2_-90 material achieves a 99.99% killing rate, whereas the BC material only reaches a 95.09% removal rate. This indicated that in a short period of the in vitro culture, due to the short contact time between the material and *E. coli*, the bacteria were not yet adsorbed onto the material surface or into the pores. However, after a longer period of the in vitro culture, relying on the material’s own physical action, it could act on the bacterial surface, causing defects and leading to cell death, thus increasing the killing rate. It further illustrated that to exert the bactericidal performance of the material, it was necessary to ensure good adsorption under electrostatic action. Moreover, the bamboo carbon material loaded with MnO_2_ at 90 °C could significantly improve its adsorption properties. Compared with the nano-aggregated particle structure of BC@MnO_2_-70 and -110, BC@MnO_2_-90 formed a well-distributed manganese oxide nanosheet structure, which can further enhance the material’s surface sharp physical cutting effect, thereby enhancing its bactericidal performance.

Figure 9 shows the FESEM images of *E. coli* after 60 min of the in vitro culture with BC and BC@MnO_2_-70, -90, and -110 materials. Figure 9a is the FESEM image of the control *E. coli*. From Figure 9b–e, it can be clearly seen that after 60 min of the in vitro culture with the bamboo carbon material, the morphology of the bacterial cells exhibited two phenomena: first, the cell surface changed from smooth to wrinkled, and second, there were obvious physical defects at both ends of the cells, with cell rupture and dissolution. Based on these observations, the mechanism of bacterial removal by bamboo carbon material during the in vitro culture is inferred. Under electrostatic action, bacterial cells were adsorbed onto the material surface, leading to a physical blocking effect. Additionally, it can be clearly seen from Figure 9b–e that after 60 min of the in vitro culture with the bamboo carbon material loaded with MnO_2_, the morphology of *E. coli* changed significantly, which was basically similar to Figure 9b–e, indicating that the bactericidal mechanisms were similar. However, the difference was that after 60 min of the in vitro culture with the bamboo carbon material loaded with MnO_2_, there were more bacteria showing signs of dissolution, and the dissolution phenomenon of BC@MnO_2_-90 was the most pronounced, indicating that the bactericidal effect of BC@MnO_2_-90 was superior to that of BC and BC@MnO_2_-70 and -110 materials. The sharp physical cutting effect caused by the manganese oxide nanosheet structure formed after loading MnO_2_ was greater than that of the bamboo carbon material.

### 3.3. CDI Process

Figure 10 shows the comparison of colony-forming units after 15 min and 180 min of CDI bacterial removal with BC and BC@MnO_2_-70, -90, and -110 materials. It is evident from the figure that there was a significant change in the number of colonies after 15 min and 180 min of CDI bacterial removal with the bamboo carbon material. Specifically, the BC@MnO_2_-90 material has fewer colonies after 180 min of CDI bacterial removal compared to BC@MnO_2_-70 and -110, reducing from 11 to 0. This indicated that the CDI capacitive bacterial removal effect of the BC@MnO_2_-90 material was the best. During the CDI capacitive bacterial removal process, the external electric field enhanced the bactericidal effect of the electrode; after 180 min, the bacteria in the water body could be essentially killed by the BC@MnO_2_-90 material, fully reflecting the rapid effectiveness of CDI capacitive bacterial removal, which could quickly enrich bacteria and achieve the purpose of killing bacteria under the action of the manganese oxide composite electrode material within a short time. At the same time, it also indicated that loading MnO_2_ can effectively enhance the surface activity of the bamboo carbon material, thereby enhancing the sharp physical cutting effect of the bamboo carbon material on bacteria.

Figure 11a,b shows the comparison of the killing rates of BC and BC@MnO_2_ materials after CDI bacterial removal for 15, 30, 60, 120, and 180 min at 0 V and 1.2 V. Comparing the results of CDI bacterial removal at 0 V and 1.2 V, it was evident that the killing rate was significantly higher when voltage was applied, indicating that the electric field produced could quickly enrich and adsorb bacteria to the anode electrode, and BC@MnO_2_-90 could achieve a 99.99% killing rate after 180 min, while the CDI bacterial removal effect at 0 V also gradually increased with time, and the bamboo carbon material loaded with MnO_2_ could also exert its own adsorption effect on bacteria and produce a certain bactericidal effect, but the effect was not ideal, with the killing rate below 92.15% after 180 min.

From Figure 11b, it can be observed that after applying 1.2 V, the killing rates of BC and BC@MnO_2_ materials after 15 min of CDI bacterial removal are all above 82.35%. After 60 min of CDI bacterial removal, the killing rates of BC@MnO_2_ materials reach 94.12%, while the killing rate of BC material was 92.15%. After 180 min of CDI bacterial removal, the killing rate of BC@MnO_2_-90 material reaches 99.99%, indicating that the CDI bacterial removal effect of BC@MnO_2_ materials was superior to that of BC material, and the CDI bacterial removal effect of BC@MnO_2_-90 material was better than that of BC and BC@MnO_2_-70 and -110 materials. It also indicated that loading MnO_2_ on the bamboo carbon material at the appropriate temperature could utilize its pseudocapacitive characteristics and microporous manganese oxide nanosheet structure to enhance the CDI capacitive bacterial removal effect.

## 4. Conclusions

The specific surface area of materials played a crucial role in electrochemical performance. Materials with larger surface area provided more active sites, which could be used for the adsorption and storage of charges, thereby increasing the charge storage capacity of the device. A larger specific surface area means that more electrode material is exposed to the electrolyte, which could reduce the distance of charge transfer and improve the reaction rate [27]. The increased surface area could provide more reaction sites and facilitate the diffusion of electrolyte ions, thereby improving the kinetics of electrochemical reactions, which was crucial for enhancing the electrochemical performance of BC@MnO_2_ samples.

Zheng discovered that a MnO_2_-based capacitive system could augment the bactericidal effect of ozone by causing damage to the bacterial cell membrane [28]. Wang’s research indicated that the Ag/MnO_2_ composite facilitated the generation of intracellular reactive oxygen species (ROS) and led to the disintegration of the cell wall and cell membrane [29]. Thus, it could be found that the sharp physical cutting bacterial removal effect of manganese dioxide mainly refers to the surface adsorption of manganese dioxide and its sharp surface structure, which may directly penetrate microbial cells, leading to mechanical damage to the bacterial cell wall shell, resulting in cell defects and cell death. In addition, manganese dioxide can to some extent induce the production of intracellular reactive oxygen species (ROS) and the rupture of the cell wall and cell membrane. This study focuses on the desalination and disinfection performance of CDI, and therefore explores the influence of the electro-adsorption capacity and specific surface area of CDI composite electrode materials on their desalination and antibacterial properties, in order to understand the performance of CDI and the antibacterial mechanism of BC@MnO_2_ composite materials [16]. In this study, the bamboo carbon and KMnO_4_ were used as raw materials, employing a one-step method to load MnO_2_ onto bamboo carbon at different temperatures (70 °C, 90 °C, 110 °C), to obtain the corresponding carbon-based composite materials. These materials were used as anodic materials for capacitive deionization (CDI) and their effects on CDI capacitive deionization performance after MnO_2_ loading are studied. It could be observed that the electrosorption capacity of the bamboo carbon material (2.34 mg/g) significantly increased after MnO_2_ loading, with the BC@MnO_2_-90 material achieving an electrosorption capacity of 4.09 mg/g. The CDI capacitive bacterial removal rate of the bamboo carbon material loaded with MnO_2_, specifically BC@MnO_2_-90, after 180 min of the culture (99.99%), was higher than that of BC@MnO_2_-70 and -110 (96.08% and 95.09%, respectively). It demonstrated that loading MnO_2_ on the bamboo carbon material at an appropriate temperature can utilize its pseudocapacitive characteristics and the microporous structure of manganese oxide nanosheets to enhance the electrosorption performance of the material under the action of the CDI electric field. It not only improved the material’s electrosorption enrichment and separation of bacteria but also enhanced the material’s sharp physical cutting bacterial removal effect, thereby further enhancing its CDI capacitive synchronous enrichment, separation, and bacterial removal performance.

## Figures and Tables

**Figure 1 nanomaterials-14-01565-f001:**
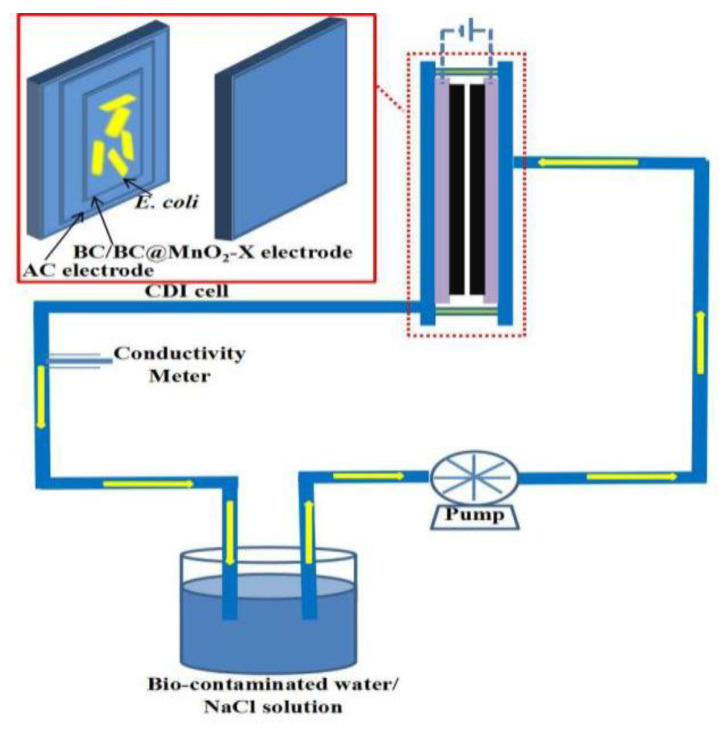
Schematic diagram of CDI device.

**Figure 2 nanomaterials-14-01565-f002:**
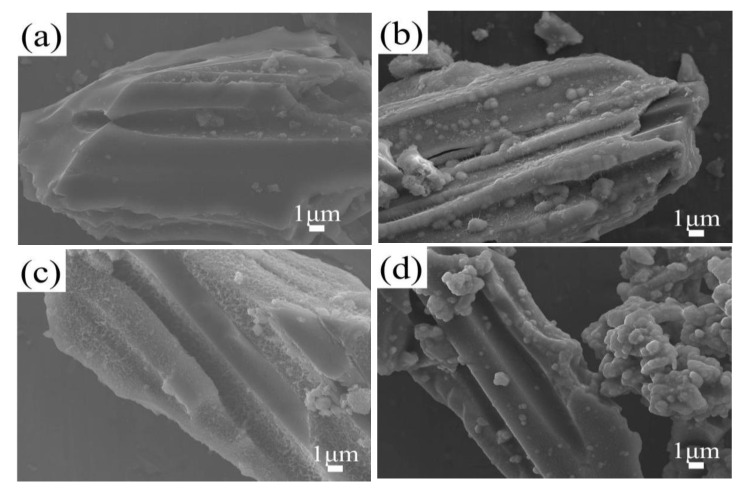
FESEM images of (**a**) BC, (**b**) BC@MnO_2_-70, (**c**) BC@MnO_2_-90, and (**d**) BC@MnO_2_-110.

**Figure 3 nanomaterials-14-01565-f003:**
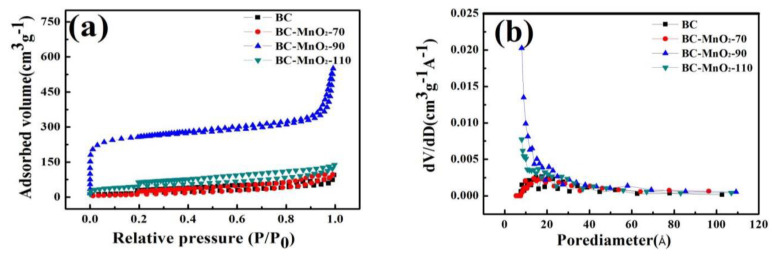
(**a**) Nitrogen adsorption–desorption isotherms and (**b**) pore size distribution of BC, BC@MnO_2_-70, BC@MnO_2_-90, and BC@MnO_2_-110.

**Figure 4 nanomaterials-14-01565-f004:**
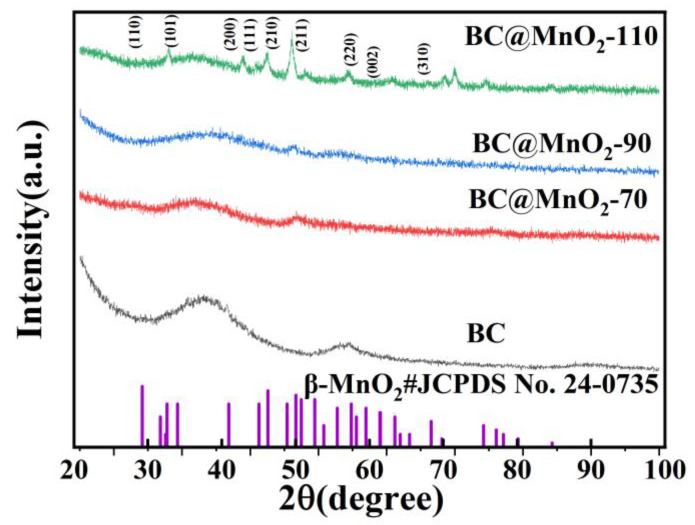
XRD patterns of BC, BC@MnO_2_-70, BC@MnO_2_-90, and BC@MnO_2_-110.

**Figure 5 nanomaterials-14-01565-f005:**
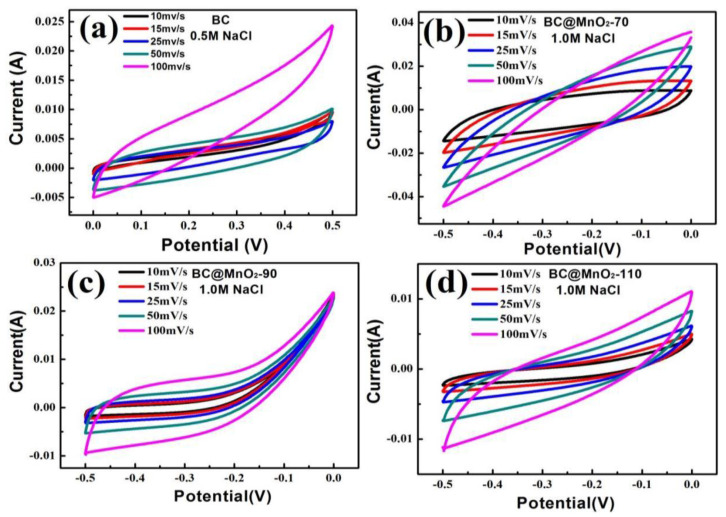
Cyclic voltammograms of (**a**) BC, (**b**) BC@MnO_2_-70, (**c**) BC@MnO_2_-90, and (**d**) BC@MnO_2_-110.

**Figure 6 nanomaterials-14-01565-f006:**
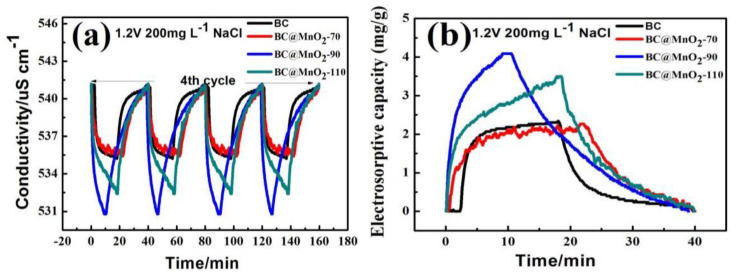
(**a**) Conductivity/time profiles of BC, BC@MnO_2_-70, BC@MnO_2_-90, and BC@MnO_2_-110 at NaCl 200 mg/L; and (**b**) electrosorptive capacity/time profiles ofBC, BC@MnO_2_-70, BC@MnO_2_-90, and BC@MnO_2_-110 at 1.2 V NaCl 200 mg/L.

**Figure 7 nanomaterials-14-01565-f007:**
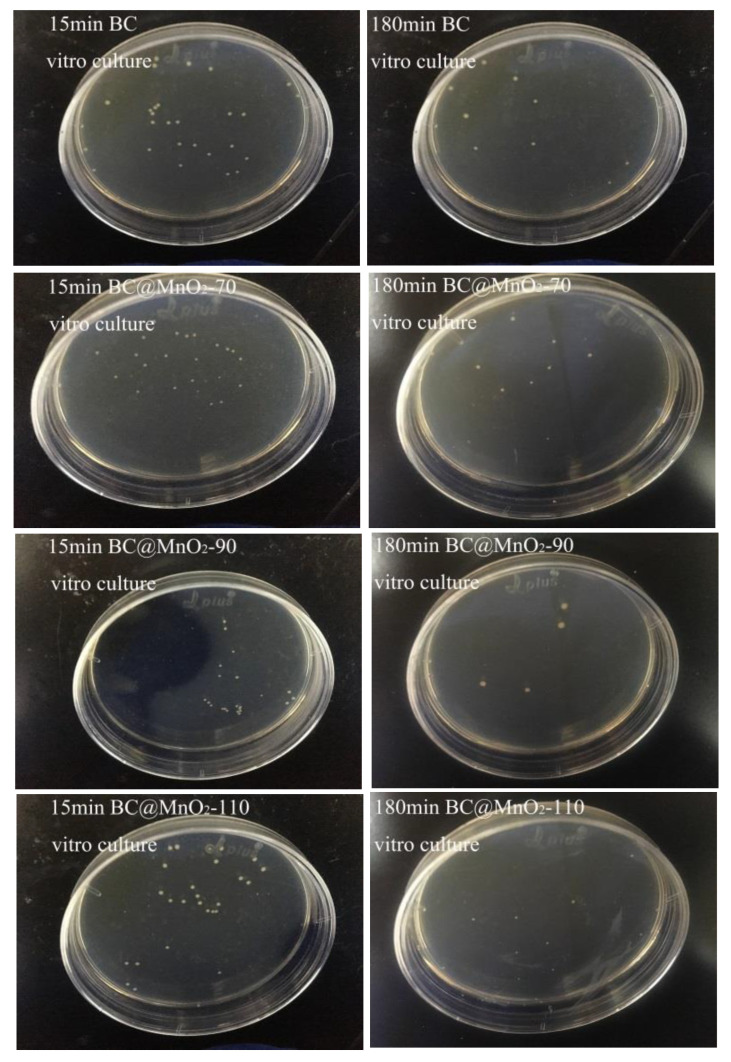
The colony-forming units after in vitro cultures using BC, BC@MnO_2_-70, BC@MnO_2_-90, and BC@MnO_2_-110 (100 µg mL^−1^) for 15 and 180 min.

**Figure 8 nanomaterials-14-01565-f008:**
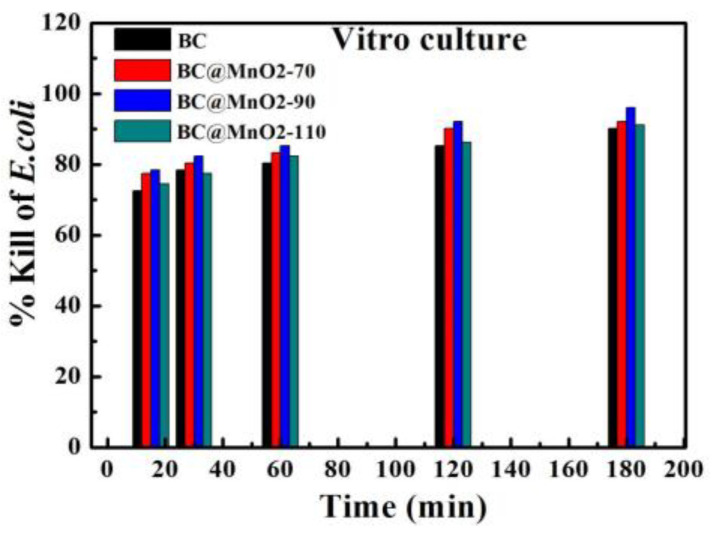
Killing rate of microbes after in vitro culture with BC, BC@MnO_2_-70, BC@MnO_2_-90, and BC@MnO_2_-110 for 15, 30, 60, 120, and 180 min.

**Figure 9 nanomaterials-14-01565-f009:**
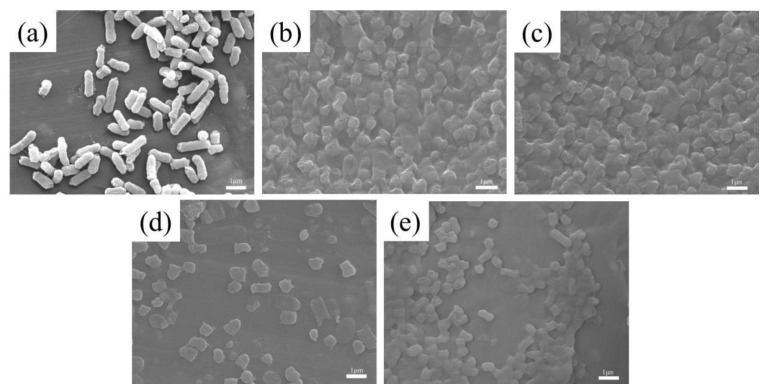
FESEM images of *E. coli*: (**a**) untreated control, and (**b**–**e**) treated with BC, BC@MnO_2_-70, BC@MnO_2_-90, and BC@MnO_2_-110 (100 µg mL^−1^) for 60 min.

**Figure 10 nanomaterials-14-01565-f010:**
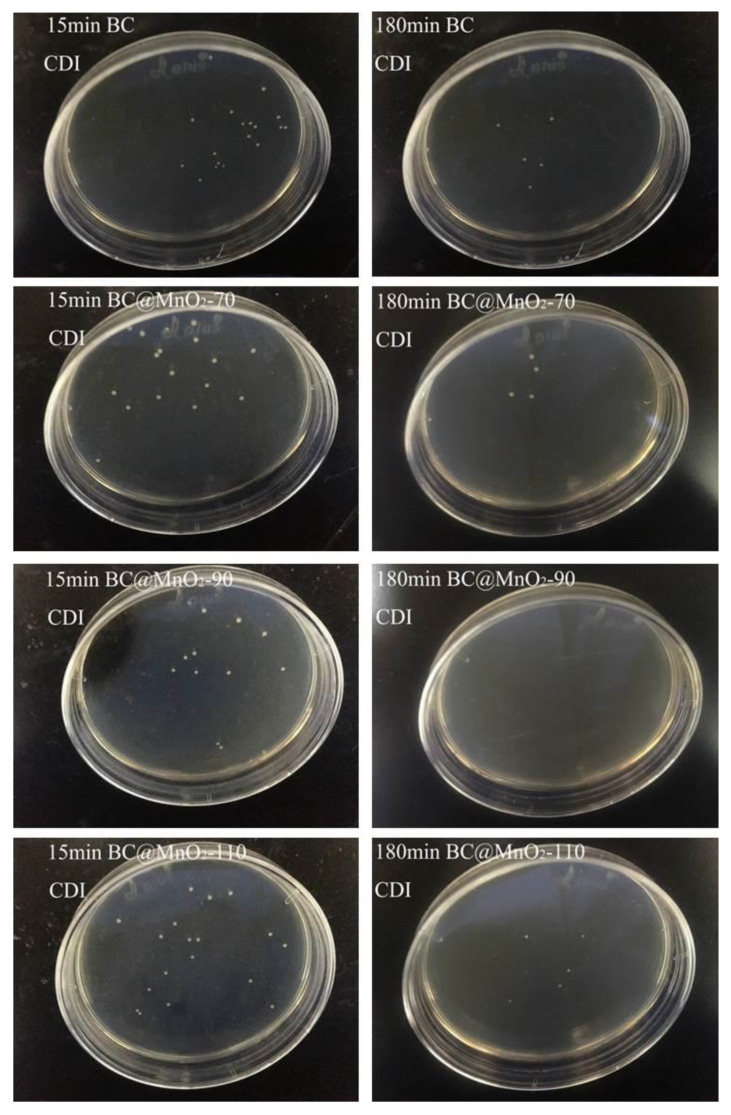
The colony-forming units after the CDI process culture with BC, BC@MnO_2_-70, BC@MnO_2_-90, and BC@MnO_2_-110 for 15 and 180 min.

**Figure 11 nanomaterials-14-01565-f011:**
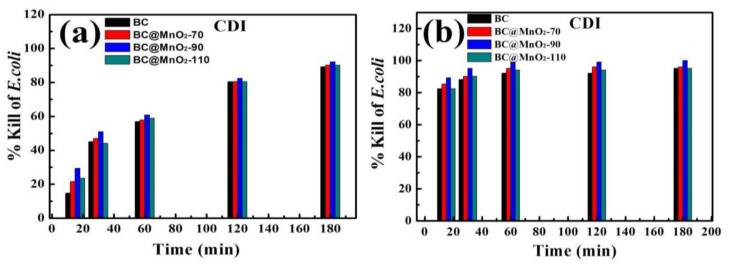
FESEM images of (**a**) BC, BC@MnO_2_-70, BC@MnO_2_-90, and BC@MnO_2_-110 electrodes after the CDI process and (**b**) the regeneration process.

**Table 1 nanomaterials-14-01565-t001:** Comparison Data from Adsorption Isotherms.

Sample	SBET (m^2^/g)	Vtot (cm^3^/g)	Dpore (Å)
BC	122.50	0.06	56.70
BC@MnO_2_-70	101.82	0.15	29.70
BC@MnO_2_-90	192.83	0.49	51.80
BC@MnO_2_-110	118.13	0.15	25.60

## Data Availability

The data presented in this research are available on request from the corresponding author.

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
