# Peer review of "Employing Manganese Dioxide and Bamboo Carbon for Capacitive Water Desalination and Disinfection"

_nanomaterials, 2024, doi:10.3390/nano14191565_

Round 1

Reviewer 1 Report

Comments and Suggestions for Authors

Dear authors

Your work, " Employing Manganese Dioxide and Bamboo Carbon for Capacitive Water Desalination and Disinfection" provides valuable insights into an effective capacitive desalination (CDI) and disinfection of water using he manganese dioxide (MnO2)/bamboo carbon (BC) composite. These carbon-based composites electrodes’ bactericidal rate for Escherichia coli could reach 99.99%. However, several points need further attention:

1)                 The authors should clearly highlight the novelty of their work in the introduction section. This will help readers understand the unique contributions of the study right from the beginning. Including this information is essential for setting the context and framing the research's importance.

2)      The XRD file reference from the JCPDS database should be included in Figure 4.and the quality of figure could be improved

3)      How can the authors explain the decrease in surface area from 192.83 m2/g for BC@MnO2-90 to 118.13 m2/g for BC@MnO2-110?

4)      How does the external electric field contribute to the enhancement of the bactericidal effect in the CDI process?

5)                 What is the primary mechanism behind the bactericidal effect observed in BC@MnO2-90 during the CDI capacitive bacterial removal process?

6)      How does the electrode polarization affect the transition from pseudocapacitance to electrolyte diffusion dominance in BC and BC@MnO2 materials?

7)      How does the porous MnO2 structure formed on the BC surface enhance Na+ adsorption on the electrode?

8)      What role does the specific surface area (SBET) of the materials play in the electrochemical performance of BC and BC@MnO2 samples?

Author Response

Comments 1: The authors should clearly highlight the novelty of their work in the introduction section. This will help readers understand the unique contributions of the study right from the beginning. Including this information is essential for setting the context and framing the research's importance.

Response 1: The novelty of the work had been clearly highlighted in the introduction section.

Comments 2: The XRD file reference from the JCPDS database should be included in Figure 4.and the quality of figure could be improved

Response 2: The XRD file reference from the JCPDS database had been added into the Figure 4.

Comments 3: How can the authors explain the decrease in surface area from 192.83 m2/g for BC@MnO2-90 to 118.13 m2/g for BC@MnO2-110?

Response 3: After loading MnO2, heat treatment is usually required to fix MnO2 on BC. The temperature and duration of heat treatment will affect the crystallinity and dispersibility of MnO2, thereby affecting the specific surface area. From SEM, it could be seen that the dispersibility of BC@MnO2-110 was worse than BC@MnO2-90. So the specific surface area of BC@MnO2-110 was lower than BC@MnO2-90.

Comments 4: How does the external electric field contribute to the enhancement of the bactericidal effect in the CDI process?

Response 4: In the process of capacitive deionization (CDI), an external electric field can increase the surface charge of the electrode material, thereby enhancing its adsorption capacity for charged microorganisms. In addition, external electric fields can also cause uneven distribution of charges on microbial cell membranes, resulting in potential differences, which may lead to electroporation or rupture of the cell membrane, thereby damaging the integrity of the cell. Moreover, the external electric field may also improve the hydrophilicity of the electrode material surface, which facilitates the contact between microbial cells and the material surface, thereby enhancing the sterilization efficiency.

Comments 5: What is the primary mechanism behind the bactericidal effect observed in BC@MnO2-90 during the CDI capacitive bacterial removal process?

Response 5: In the CDI process, the surface charge of the electrode material attracted and captured microorganisms with opposite charges, causing a change in the potential of the microbial cell membrane and affecting the normal function of the cell. Moreover, electric fields may cause ion migration inside and outside microbial cells, leading to an imbalance in ion concentration inside and outside the cells and affecting their normal function. In addition, the surface adsorption and sharp surface structure of the electrode material manganese dioxide may directly penetrate microbial cells, causing mechanical damage to the bacterial cell wall shell, leading to cell defects and cell death. And MnO2 had strong oxidizing properties and could generate highly reactive free radicals, which could damage bacterial cell walls and membranes, leading to bacterial cell damage and death.

Comments 6: How does the electrode polarization affect the transition from pseudocapacitance to electrolyte diffusion dominance in BC and BC@MnO2 materials?

Response 6: As the charge discharge rate increased, the polarization degree of the electrode also increased, which leaded to more charge storage processes transitioning from fast pseudo capacitive behavior to slower electrolyte diffusion processes. And the high polarization of the electrode may also cause blockage of ion transport channels, thereby reducing pseudo capacitance and increasing electrolyte diffusion.

Comments 7: How does the porous MnO2 structure formed on the BC surface enhance Na+ adsorption on the electrode?

Response 7: MnO2 provided a larger specific surface area, which means there were more active sites available for Na+ ion adsorption. The porous structure of MnO2 helped to form a more uniform charge distribution on the electrode surface and provided more ion transport channels, enabling Na+ ions to diffuse more effectively into the interior of the electrode material, thereby enhancing the adsorption force for Na+ ion. MnO2 was usually more hydrophilic than activated carbon, and this hydrophilicity helped hydrated Na+ ions to more easily approach and adsorb onto the electrode surface.

Comments 8: What role does the specific surface area (SBET) of the materials play in the electrochemical performance of BC and BC@MnO2 samples?

Response 8: The specific surface area of materials played a crucial role in electrochemical performance. Materials with larger surface area provided more active sites, which could be used for adsorption and storage of charges, thereby increasing the charge storage capacity of the device. A larger specific surface area means more electrode material is exposed to the electrolyte, which could reduce the distance of charge transfer and improve reaction rate. The increased surface area could provide more reaction sites and facilitate the diffusion of electrolyte ions, thereby improving the kinetics of electrochemical reactions, which was crucial for enhancing the electrochemical performance of BC@MnO2 samples.

Reviewer 2 Report

Comments and Suggestions for Authors

The following points are recommended for the authors to address:

1. Line 113 “The bio-contaminated water/NaCl was pumped into the CDI device at a flow rate of 12 mL/min under 1.2 V.” – Please outline the concentrations and properties of the feed solution. Also, it is essential to mention what type of water (e.g., seawater, brine, or wastewater etc) the feed solution is simulating or mimicking. This is very important for readers to understand the need for this work.

2. Line 76 – Housekeeping matters, some digits in chemical formulations are not in subscript form.

3. Line 15 “these composites electrodes exhibited good cycle stability, electrosorption capacity (4.09 mg/g) and excellent bactericidal effect” – Please specify the target ion/material that the electrode is adsorbing. i.e., 1 of adsorbent could capture 4.09 mg of xx?

4. Line 35 “Characterized by its non-toxicity and absence of pollution, CDI technology offers a simple and cost-effective operation” – In our opinion, it is essential to mention the merits of CDI as compared to surface-based physicochemical methods. Faster kinetics is one key merit. Secondly, the presence of an electric field could accelerate the diffusion of ions, making it possible to overcome the adsorption-desorption equilibrium encountered in physicochemical adsorption that is controlled by thermodynamics (Ref: J. Mater. Chem. A, 2023,11, 22551-22589). These points are important to mention in the context of this paper and we suggest the authors to discuss them in the manuscript.

5. Since the goal of this work is to examine new materials for capacitive desalination (NaCl was used) and bactericidal effects, it seems that the application of this work is geared towards seawater desalination or wastewater treatment where NaCl is present, and thus anti-biofouling properties are needed due to the high fouling potential of these waters (please see Ref. in previous comment). It is suggested for the authors to add a discussion on this so that readers can better appreciate the context of this work.

6. Line 378 – Please define “sharp physical cutting bacterial removal effect,” at first mention. Readers may not be familiar with this term.

7. Line 335 “the bamboo carbon material loaded with MnO2 could also exert its own adsorption effect on bacteria and produce a certain bactericidal effect, but the effect was not ideal, with the killing rate below 92.15% after 180 minutes.” – Please outline the significance of 99.99% kill rate. As compared to 92.15%, how important is the additional 7% kill rate? Would this value be significant for a particular application? Please explain.

Author Response

Comments 1: Line 113 “The bio-contaminated water/NaCl was pumped into the CDI device at a flow rate of 12 mL/min under 1.2 V.” – Please outline the concentrations and properties of the feed solution. Also, it is essential to mention what type of water (e.g., seawater, brine, or wastewater etc) the feed solution is simulating or mimicking. This is very important for readers to understand the need for this work.

Response 1: The concentration and properties of the feed solution are bio-contaminated water (106 CFU mL-1 suspension of Escherichia coli)/NaCl (200mg·L-1). The feed solution simulates or imitates bio-contaminated water with a certain concentration of Escherichia coli or saltwater.

Comments 2: Line 76 – Housekeeping matters, some digits in chemical formulations are not in subscript form.

Response 2: Modified and annotated according to the format requirements

Comments 3: Line 15 “these composites electrodes exhibited good cycle stability, electrosorption capacity (4.09 mg/g) and excellent bactericidal effect” – Please specify the target ion/material that the electrode is adsorbing. i.e., 1 of adsorbent could capture 4.09 mg of xx?

Response 3: The target ion for electrode adsorption has been designated as Na+ ion.

Comments 4: Line 35 “Characterized by its non-toxicity and absence of pollution, CDI technology offers a simple and cost-effective operation” – In our opinion, it is essential to mention the merits of CDI as compared to surface-based physicochemical methods. Faster kinetics is one key merit. Secondly, the presence of an electric field could accelerate the diffusion of ions, making it possible to overcome the adsorption-desorption equilibrium encountered in physicochemical adsorption that is controlled by thermodynamics (Ref: J. Mater. Chem. A, 2023,11, 22551-22589). These points are important to mention in the context of this paper and we suggest the authors to discuss them in the manuscript.

Response 4:These points had been added to this study..

Comments 5:  Since the goal of this work is to examine new materials for capacitive desalination (NaCl was used) and bactericidal effects, it seems that the application of this work is geared towards seawater desalination or wastewater treatment where NaCl is present, and thus anti-biofouling properties are needed due to the high fouling potential of these waters (please see Ref. in previous comment). It is suggested for the authors to add a discussion on this so that readers can better appreciate the context of this work.

Response 5: These points had been added to the a discussion.

Comments 6:  Line 378 – Please define “sharp physical cutting bacterial removal effect,” at first mention. Readers may not be familiar with this term.

Response 6: The definition of “sharp physical cutting bacterial removal effect” had been added to the discussion.

Comments 7:  Line 335 “the bamboo carbon material loaded with MnO2 could also exert its own adsorption effect on bacteria and produce a certain bactericidal effect, but the effect was not ideal, with the killing rate below 92.15% after 180 minutes.” – Please outline the significance of 99.99% kill rate. As compared to 92.15%, how important is the additional 7% kill rate? Would this value be significant for a particular application? Please explain.

Response 7: The 99.99% kill rate means that almost all target microorganisms are killed, with only a very small number possibly surviving. Compared to a kill rate of 92.15%, an additional 7.84% may mean less microbial survival, which is crucial in controlling infection risks, ensuring product quality, and safety. For capacitive deionization (CDI) disinfection, a 99.99% sterilization rate may be equally important. In the water treatment process, a high disinfection rate helps ensure that the water quality meets safety standards, especially in applications that require high-purity water.

Round 2

Reviewer 2 Report

Comments and Suggestions for Authors

Accept. The authors have addressed the comments.